# Improved Radar Composites and Enhanced Value of Meteorological Radar Data Using Different Quality Indices

**Ladislav Méri [1,2], Ladislav Gaál [3,\*], Juraj Bartok [2,3], Martin Gažák [3], Martin Gera [2], Marián Jurašek [1] and Miroslav Kelemen [4]**

[1] Remote Sensing Department, Slovak Hydrometeorological Institute, 833 15 Bratislava, Slovakia; ladislav.meri@shmu.sk (L.M.); marian.jurasek@shmu.sk (M.J.)

[2] Department of Astronomy, Physics of the Earth, and Meteorology, Comenius University in Bratislava, 842 48 Bratislava, Slovakia; juraj.bartok@microstep-mis.com (J.B.); martin.gera@fmph.uniba.sk (M.G.)

[3] MicroStep-MIS, Čavojského 1, 841 04 Bratislava, Slovakia; martin.gazak@microstep-mis.com

[4] Faculty of Aeronautics, Technical University of Košice, 041 21 Košice, Slovakia; miroslav.kelemen@tuke.sk

\* Correspondence: ladislav.gaal@microstep-mis.com

**Abstract:** Radar measurements are inherently affected by various meteorological and non-meteorological factors that may lead to a degradation of their quality, and the unwanted effects are also transferred into composites, i.e., overlapping images from different radars. The paper was aimed at answering the research question whether we could create 'cleaner' radar composites without disturbing features, and if yes, how the operational practice could take advantage of the improved results. To achieve these goals, the *qRad* and *qPrec* software packages, based on the concept of quality indices, were used. The *qRad* package estimates the true quality of the C-band radar volume data using various quality indices and attempts to correct some of the adverse effects on the measurements. The *qPrec* package uses a probabilistic approach to estimate precipitation intensity, based on heterogeneous input data and quality-based outputs of the *qRad* software. The advantages of the *qRad* software are improved radar composites, which offer benefits, among others, for aviation meteorology. At the same time, the advantages of the *qPrec* software are manifested through improved quantitative precipitation estimation, which can be translated into hydrological modeling or climatological precipitation mapping. Beyond this, the developed software indirectly contributes to sustainability and environmental protection—for instance, by enabling fuel savings due to the more effective planning of flight routes or avoiding runway excursions due to information on the increased risk of aquaplaning.

**Keywords:** radar meteorology; radar composite; quality index; quantitative precipitation estimation; aviation safety

## 1. Introduction

Meteorological radars are an important part of the everyday practice of weather and aviation professionals and have a direct impact on public welfare. They provide valuable information on the hydrometeor-related atmospheric processes and cover various spatial and temporal scales: the movement of weather systems, genesis/evolution/decomposition of cloud systems, rainfall rates, form and phase of hydrometeors, etc. [1–3], and all of these are essential inputs in forecasting/nowcasting models [4]. One of the most significant drawbacks of radar-based observations is the questionable precision of the rainfall amount estimation, in comparison with the true amounts measured by a ground-based network of rain gauges [5]. Practically, one of the most essential goals of radar meteorology is adjusting the inherently imprecise radar measurements to the spatially relatively sparse gauge-based information on rainfall [6].

More radars (i.e., an increase in quantity) do not necessarily imply better rainfall estimates (i.e., an increase in quality). Nevertheless, with correctly chosen approaches to

creating radar composites (i.e., overlapping images from different radars), one can also enhance the quality of the radar information [7–9]. Homogeneous radar composites over larger areas or locally in complex terrain, created from a heterogeneous radar network, provide up-to-date information on meteorological processes in the lower air-space. Observations and, more importantly, nowcasting of thunderstorms allow for the strategic planning of airport operations and flight routes. Increased situational awareness contributes to safety, particularly during landing or take-off, for instance in minimizing the risk of runway excursions. These are the most frequent type of runway safety accidents—they add up to 25% of all accidents over the 2015–2019 period according to 2019 Safety Report of the International Air Transport Association [10]. Additionally, preparedness and ability to cope with unexpected situations and/or adverse weather events [11] results in optimized operating schedules, which save fuel, reduce delays, minimize diversions (e.g., [12]), and, thus, reduce costs and decrease the greenhouse gas emissions. The improved radar composites may, therefore, indirectly increase efficiency and enhance sustainability of the air traffic management.

High quality radar information is particularly preferred in aviation; thus, aircraft are currently compulsorily equipped with airborne radars. These instruments, however, have limited power, and therefore, are only able to cover a shorter range ahead of the aircraft. Radar composites, on the other hand, offer more detailed information regarding the meteorological conditions within a much wider range along the flight route and in the vicinity of the target airport.

There are a number of verified and approved methods for constructing radar composites in the scientific literature, either on the level of cases studies or in the form of implemented algorithms from the operative practice of hydrometeorological services. Nevertheless, radar manufacturers generally do not devote dedicated attention to the issue of compositing, as they primarily focus on the single-radar algorithms. This is, on one hand, logical, since well-functioning single radars are the necessary condition of any radar networking. On the other hand, this is a topic that requires a more elaborate approach mainly in the light of increased demands for radar data within pan-European data exchange and co-operation [7–9].

One of the basic dilemmas when evaluating data from more than one radars ($N > 1$) at a given point (pixel, bin) is how to treat the multiplicity of the reflectivity information $Z_i$, $i = 1, \ldots, N$. There are different, generally accepted methods: to take the maximum value of the available reflectivity $Z_i$, to estimate the mean reflectivity, or to compute some weighted average of the $Z_i$ values. In the latter case, the definition of the appropriate weights offers plenty of further alternatives to cope with the problem: for instance, with the weighting factors being inversely proportional to the distance of the given pixel from each radar, with the inclusion of the actual height of the radar beam above the ground within the target pixel, or with the combination of these or further variables. Jurczyk et al. [13] presented a substantial overview of the pros and cons of different compositing methods and emphasized that the weights might also express the quality of the data from the contributing radars.

The quality of the radar measurements may be expressed through quality information or quality indices (QI, [14]). Beyond the meteorological targets and its effects (e.g., bright band), the radar signal is affected by a number of further factors stemming from the technological construction of the radar itself, the geomorphological character of the surrounding area (e.g., ground clutter, beam blocking), different non-meteorological targets (e.g., birds, insects, interfering wi-fi signals), the parameter settings of the particular radar scan (height of the radar beam above the target points, the age of the reflectivity information within a given scan), etc. The effect of all these factors may be converted into mathematical form as quality indices, and the combination of these phenomena-related, particular QIs might be understood as the overall quality index of the given radar [15]. There is no general convention regarding how to define and/or how to use the different QIs—their usage depends on

the goals of the research teams or the operational requirements of the hydrometeorological services [13,16–23].

A common measure of the quality of the QI-based estimation is some visual improvement in the layout of the radar composites. Visual impressions are subjective, and they are hard to quantify; thus, the benefits of the QI-based approaches are usually evaluated indirectly, through the improvement of the performance of various statistical, weather forecasting, or hydrological models. Such a category of statistical models is quantitative precipitation estimation (QPE). It is beyond the goals of the current study to present an overview of the methods of radar-based QPE [4]; however, we can mention a couple of works especially devoted to QI-based QPE. Szturc et al. [20,21] developed a method of QPE in a probabilistic framework. Their concept assumes that the probability density function (PDF) of the rainfall amounts (rates) can be approximated by the two-parameter gamma distribution, where the parameters of the PDF are functions of the radar deterministic measurement $R$ and the quality index QI. With the estimated PDF parameters for each pixel, ensembles of precipitation fields can be generated, which further serve as inputs for hydrological modeling [21]. Zhang et al. [24] developed QI-based QPE for the national network of polarimetric radars in China with QIs based mostly on polarimetric parameters.

The current paper was motivated by the development of two inter-related software packages, *qRad* and *qPrec*, that were built to take advantage of the quality information from the available C-band radar measurements. The software packages follow the recent trends in the development of radar applications and are in line with the recommendations of the OPERA program (Operational Programme for the Exchange of Weather Radar Information) of the EUMETNET (Network of European Meteorological Services) [9]. The developed software can indicate noticeable results in the fields of operative meteorological or hydrological practice, with the potential for use in aviation meteorology. The benefits of the *qRad* software are represented by improved radar composites to better aid air navigation and safety, whereas those of the *qPrec* software are manifested through enhanced QPE, which than can be translated into more realistic hydrographs as outputs from hydrological models, and the estimation/forecast of runway conditions in air traffic management.

## 2. Methods

The development of the *qRad* radar data processing and the *qPrec* precipitation estimating software started in 2015 at the Slovak Hydrometeorological Institute (SHMU). Both packages are based on a network-centered concept where the data from different radars are not processed independently, but simultaneously, together with further data sources. The data from the available radar network are inter-compared, controlled, and combined with each other. The software is written in the C++ language in a modular way that allows for easy portability, and uses parallel processing techniques to reduce the computing time.

The *qRad* software assesses the actual quality of the radar volume data by means of various quality indices and aims to correct some undesirable effects of the measurement (e.g., beam-blockage, ground-clutter, and non-meteorological echoes). The corrected radar volume data are then directly processed to composite radar products and quality maps. Sections 2.1 and 2.2 below provide a description of the concept of the radar quality assessment.

The *qPrec* package adopts estimation theory and a probabilistic approach to estimate the rainfall intensity as accurately as possible, based on heterogeneous input data (rain-gauges, radars, satellites, the potential of using lightning detection in the future, etc.). The calibrated input data fields are combined according to their precision and quality, resulting from the analysis of the *qRad* software. In other words, the quality indices of the radar measurements represent the link that couples the *qPrec* and the *qRad* software. The concept of the *qPrec* software will be presented in more detail in Section 2.3.

### 2.1. Specific Measures of the Radar Data Quality

Any measurement of a meteorological radar is influenced by a variety of factors. Some of these originate from the radar hardware and the settings of the measurement parameters (e.g., the power, polarity, beam-width, noise, detection range, and various filters on the hardware side). Other factors are caused by the surrounding environment (e.g., beam height above the terrain, beam-blockage, sea-clutter, and interference with other electronic devices) or by the actual meteorological situation (e.g., beam propagation, partial beam-filling, bright-band, overshooting of the low clouds, and second-trip echoes). The effect of the majority of these quality factors can be expressed in a mathematical form, or we can estimate their probability. The most convenient way of using the quality factors (quality indices, QI) is in a form of dimensionless numbers between 0.0 and 1.0, where the value 0.0 (1.0) stands for the lowest (highest) quality. The *qRad* software estimates the quality of each measured bin in the radar volume data using different quality indices.

The current version of the *qRad* software is based on a combination of nine quality indices. These evaluate the following factors:

- The distance from the radar;
- The beam height above the terrain;
- The beam blockage by the terrain;
- The similarity of the target bin to the surrounding ones;
- The time quality;
- The cloud type quality;
- The cloud top height;
- The average quality;
- The constant quality.

The listed QIs can be divided into two groups based on the principle of whether the same or a similar concept of the given QI has already appeared relatively frequently in previous studies (the first five QIs above), or the use of the particular QI is less frequent and/or it represents a novel concept of the authors (the last four QIs). The definition, the mathematical formulation (if extant) and the conceptual background of the particular QIs together with the references to formerly published studies are summarized in Appendix A. Here, we provide a formal definition of the individual QIs in the form of shadowed masks in Figure 1 with short comments to them in the following paragraph.

An example of a PPI scan (plan position indicator) made by the radar located in Brzuchania (south of Poland) is shown (Figure 1a) along with the masks of all the QIs discussed herein (Figure 1b–j). The distance QI that expresses the degradation of the radar signal quality with the distance is clearly radially symmetrical (Figure 1b). The beam height above the terrain has an important role in rainfall estimation: the closer the target bin to the surface, the more precise the estimation. In Figure 1c, the beam height QI is of an acceptable quality only within a radius of about 100 km from the radar. The beam blockage QI (Figure 1d) apparently reflects the roughness of the surrounding terrain that is characteristic for the particular radar. The similarity QI evaluates the similarity of the target bin with those within a 3 ×3 window centered on it. As demonstrated by Figure 1e, the similarity QI practically copies the edges of the objects detected by the radar.

The time QI expresses the temporal degradation of the radar signal within a single antenna rotation. Figure 1f indicates that the most recent information on the scan (i.e., higher values of the time QI) comes from the north-to-north-eastern sector, while the largest time differences (i.e., lower time QI) are assigned to the bins in the north-west-to-northern sector. The dark spots on the mask of the cloud type QI (Figure 1g) indicate echoes where no clouds were identified by the NWC SAF product (Nowcasting Satellite Application Facilities, [25]); however, radar detected some backscattered signal, perhaps due to non-meteorological targets, electromagnetic disturbance, or ground clutter. The cloud top height QI (Figure 1h) conveys similar information but with the focus on the altitudes over the clouds. The average QI (Figure 1i) shows the average value of the two previous cloud-based QIs for the past 1-h interval, indicating the locations that are

permanently affected by non-meteorological echoes. Finally, the value of the constant QI (Figure 1j) is chosen based on consideration on the degree of other disturbances affecting the radar signal, on the temporal and spatial resolution of the given radar, or on its scanning regime. For instance, the first elevation of the Polish radars are generally high (0.5°); thus, their beams are relatively high over the territory of Slovakia, and therefore they are less valuable for the Slovakia-focused radar composites.

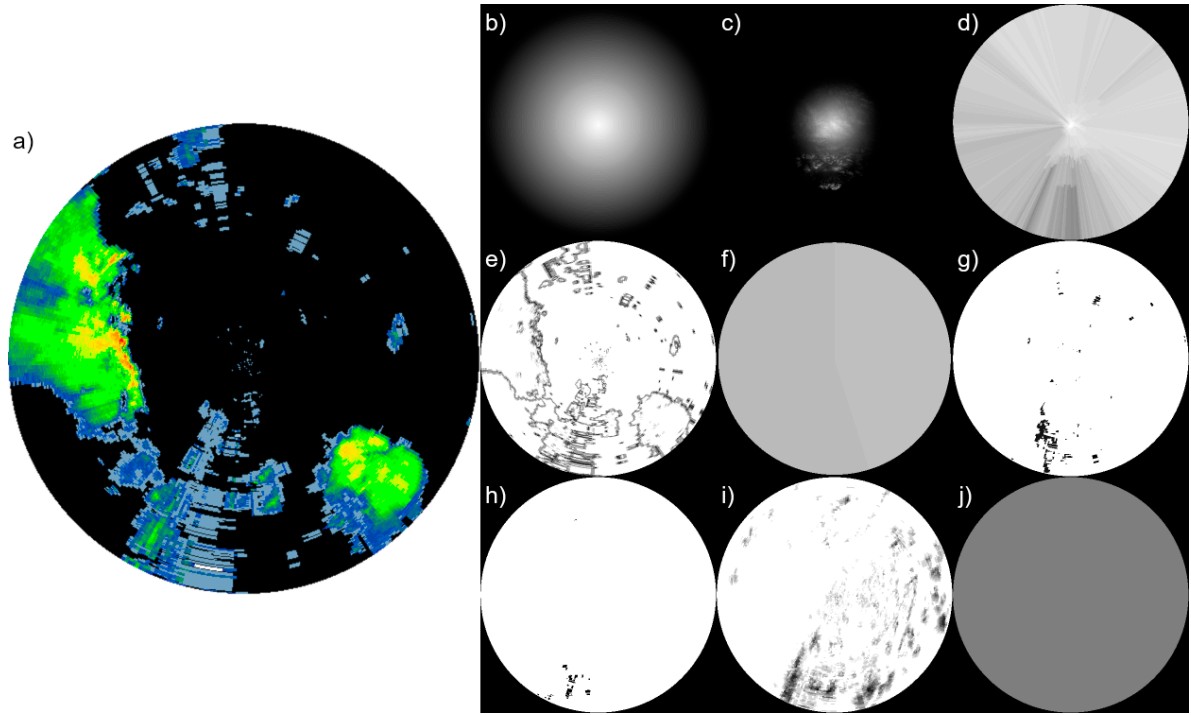

**Figure 1.** A selected PPI (plan position indicator) scan (**a**) of the radar located at Brzuchania, south of Poland (50°23′10″ N 20°5′41″ E) from 7 June 2020, 22:00 UTC, and the corresponding masks of the individual quality indices: (**b**) distance QI, (**c**) beam height QI, (**d**) beam blockage QI, (**e**) similarity QI, (**f**) time QI, (**g**) cloud type QI, (**h**) cloud top height QI, (**i**) average QI, and (**j**) constant QI. The white color of the masks indicates perfect quality (*qi* = 1.0), whereas the black color indicates poor quality (*qi* = 0.0). The range of the radar is 250 km.

The complexity of the quality assessment of the radar measurements is also illustrated by the fact that, beyond the discussed nine QIs herein, there is a wide variety of further effects that can be (and have been) quantified via different QIs. Friedrich et al. [19] attempted to approximate the vertical reflectivity profile with an enhanced focus on the bright band. Unique QIs that appeared only in the study of Fornasiero et al. [16] were those related to vertical continuity and the antenna pointing error. Szturc et al. [20,21] made use of QIs related to the number and quality of the precipitation rate products that were utilized within the process of the quantitative precipitation estimation.

More generally, one may practically quantify any phenomena that affects the propagation of the radar beam, and express it in the form of a corresponding QI. The only limitation is the meaningfulness of the defined QIs and their practical usability. The topic of QIs is also being intensively discussed within the OPERA program of the EUMETNET, where the first author of the current paper is also involved as a correspondent for the SHMU [26–28].

### 2.2. The Overall Measure of the Radar Data Quality

The overall QI of each radar bin is estimated in a multiplicative form, using the individual specific QIs (Appendix A):

$$qi_i = \prod_{j=1}^{n} qi_{j,i} \tag{1}$$

where $qi_i$ is the overall QI of the target bin from the $i$-th radar, $qi_{j,i}$ is the $j$-th specific QI of the target bin from the $i$-th radar, and $n$ is the total number of the specific QIs. The final value of the radar product is then estimated as a QI-weighted average from all available radars at the given point:

$$z = \frac{\sum_{i=1}^{N} qi_i z_i}{\sum_{i=1}^{N} qi_i} \tag{2}$$

where $z$ is the value of the resulting radar product, $z_i$ is the corresponding value from the $i$-th radar, and $N$ is the number of radars.

Using a probabilistic approach, one can estimate the overall QI of all radars at the given point. This is simply expressed as 1.0 minus the probability of a bad measurement at the given point:

$$qi = 1.0 - P' = 1.0 - \prod_{i=1}^{N} P'_i = 1.0 - \prod_{i=1}^{N} (1.0 - qi_i) \tag{3}$$

where $P'$ and $P'_i$ are the probabilities of bad measurement for the resulting product and for the $i$-th radar, respectively.

The method of using the resulting QI also depends on the target radar product. Currently, no volume filtering is adopted, i.e., the individual specific QIs are combined into the overall QI according to Equations (1) and (3). We could, in principle, introduce volume filtering by defining "no-data" in the case of QI not exceeding a pre-defined threshold. Nevertheless, different methods of combination of the specific QIs and the volume filtration methods have not been studied so far and need to be tested in the light of the chosen target radar product.

### 2.3. The QI-Based Estimation of Precipitation

The design and development of the *qPrec* precipitation estimation software was started from the scratch. The governing idea was to combine probabilistic and variational approaches. The schematic of the processing chain of the *qPrec* software is sketched in Figure 2.

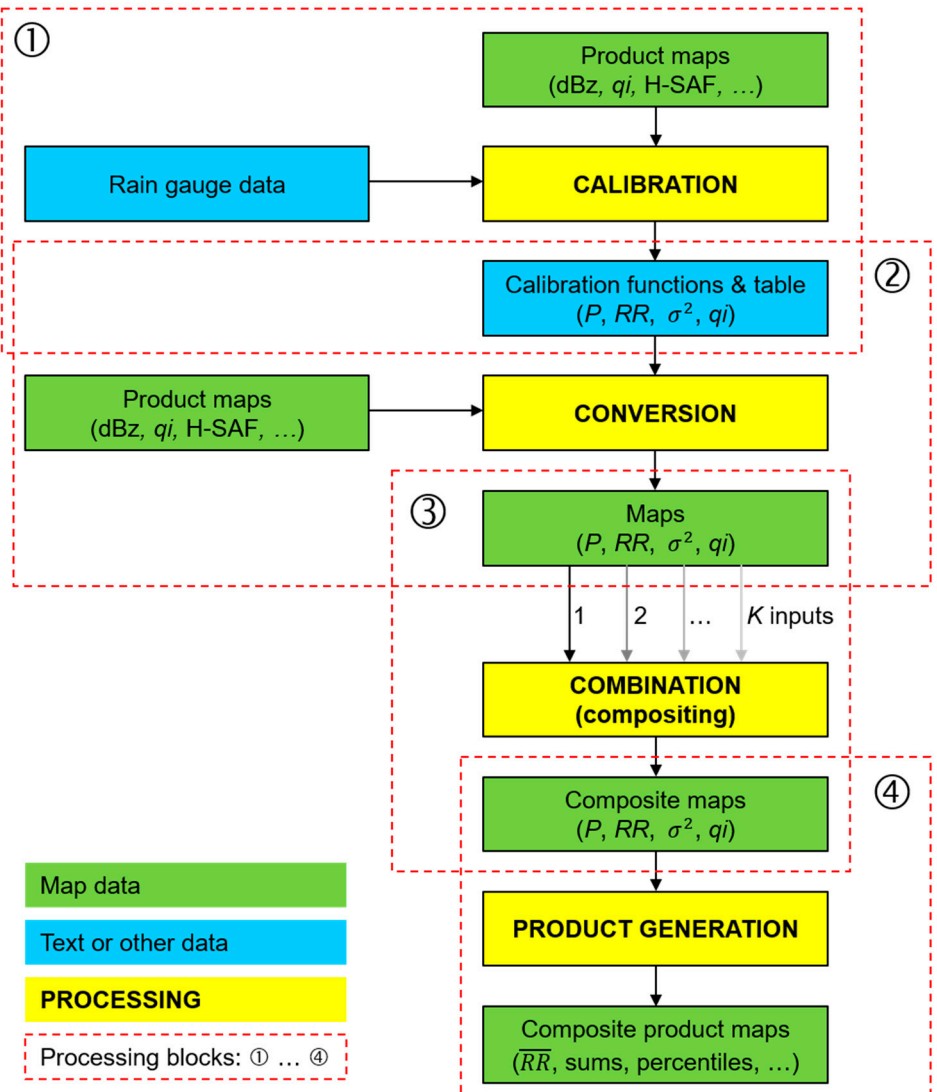

**Figure 2.** The processing chain of the *qPrec* software. "dBz" indicates the radar reflectivity products, *qi* stands for quality indices, and H-SAF is the EUMETSAT Satellite Application Facility on Support to Operational Hydrology and Water Management. *K*—the total number of the rainfall-related products, *P*—the probability of precipitation, *RR*—the rainfall rate, $\sigma^2$—the variance of the rainfall rate.

The *qPrec* processing chain (Figure 2) consists of four blocks. The first one is the calibration step. At the very beginning, the precipitation-related variables are selected. This may be, for instance, the radar reflectivity in terms of the CMAX (column max, i.e., the highest reflectivity in the corresponding column above the Earth's surface), CAPPI (constant altitude PPI, i.e., a slice through a multitude of different PPI scans at a predefined altitude above the Earth's surface), etc.; further *qi*, H-SAF (EUMETSAT Satellite Application Facility on Support to Operational Hydrology and Water Management [29]; for instance, the H-SAF product P-IN-SEVIRI (H03B)—instantaneous maps by infrared images from operational geostationary satellites "calibrated" by precipitation measurements from satellite sensors in sun-synchronous orbits); or any other variable that is directly associated with the precipitation, such as the lightning density.

Let us denote the selected precipitation-related variables as $y_k$, $k = 1, \ldots K$, where $K = N + M$, *N* is the total number of radars at the given point, and *M* is the number of additional, rainfall-related fields (such as an H-SAF product as mentioned above). It is not advised to select different radar reflectivity products at the same time (such as CAPPI and CMAX), due to their cross-correlation.

The input fields of $y_k$ are calibrated by the rain-gauge data to obtain an unbiased estimate of the rainfall-rate derived from the given input variable. The calibration is carried out on the basis of pairs of the rain gauge measurements and the corresponding radar-based rainfall estimation within the range of the given radar (usually approximately for 100–200 rain gauges) and for a long-term period lasting for several weeks (which, for instance, allows retaining the seasonal or intra-annual variability of the precipitation).

The outputs of the first processing block are four functions that describe the actual relationship between the input field $y_k$ on one hand, and the probability of precipitation $P$, the mean rainfall rate $RR$, the variance of the rainfall rate $\sigma^2$, and the mean quality index $\overline{qi}$ of the inputs on the other ($f_P$, $f_{RR}$, $f_\sigma$ and $f_{qi}$, respectively). Figure 3 presents an example of these four relationships for CMAX as the precipitation-related input field.

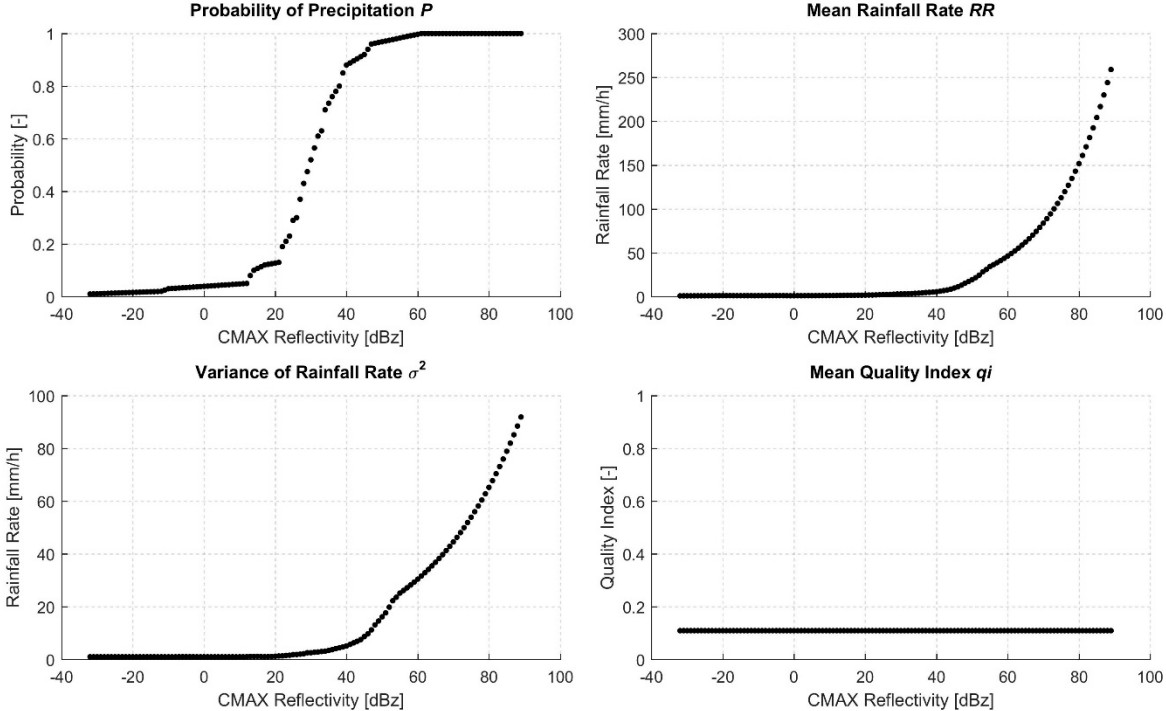

**Figure 3.** Example of the calibration functions for the CMAX (column max) product from the SHMU radar at Malý Javorník (7 June 2020).

The second processing block is the conversion. In this step, the input maps are converted by the calibration functions to maps of the mean rainfall rate, variance of the rainfall rate, probability of precipitation, and quality index.

The processing Blocks #1 and #2 are repeated $K$ times with each circle corresponding to a different input field $y_k$, $k = 1, \ldots, K$.

The third block is the combination of the $K$ converted input maps into a common map of the mean rainfall rate, variance of the rainfall rate, probability of precipitation, and quality index. The second and third block can be characterized by Equations (4)–(7):

$$P_c = \frac{\sum_{k=1}^{K} qi_k \, f_P(y_k)}{\sum_{k=1}^{K} qi_k} \tag{4}$$

$$RR_c = \frac{\sum_{k=1}^{K} f_{RR}(y_k)/\sigma_k^2}{\sum_{k=1}^{K} 1/\sigma_k^2} \tag{5}$$

$$\sigma_c^2 = \frac{1}{\sum_{k=1}^{K} 1/\sigma_k^2} \tag{6}$$

$$\sigma_k = \frac{\overline{qi}}{qi_k} f_\sigma(y_k) \tag{7}$$

where $P_c$, $RR_c$, and $\sigma_c^2$ are the resulting combined (or 'composite', hence the index 'c') probability of the precipitation, mean rainfall rate, and variance of the rainfall rate, respectively. $y_k$ is the $k$-th input value at the given point, and $qi_k$ is the quality of the $k$-th input value at the given point (i.e., overall or combined quality index, Equation (1). $f_P$, $f_{RR}$, and $f_\sigma$ are the calibration functions for the probability of the precipitation, the mean rainfall rate and the variance of the rainfall rate, and $\overline{qi}$ is the mean quality index of the inputs used in the calibration step. As Equation (7) indicates, the actual QI is used to scale the variance of the actual input value, which is then used as the weighting factor in Equations (5) and (6).

The last processing block is the product generation. Given the probability of zero precipitation ($1-P_c$) and the parameters of the non-zero rainfall rate distribution ($RR_c$, $\sigma_c^2$), the probability density function of the rainfall rate at the given point is known, and several products can be generated—for instance, the actual rainfall rate ($P_c \cdot RR_c$), and the probability of rainfall rate levels, percentiles, sums, etc.

## 3. Results

It is generally not straightforward to quantify the degree of improvement of the radar composites objectively, in light of the novel QI algorithms. Beyond the visual analysis of the radar composites, it is difficult to find any method that can directly evaluate the effect of the innovative methods in their preparation. Obviously, there is a number of indirect evaluation methods, for instance through the outputs of numerical weather prediction models or hydrological models—these will be discussed in the upcoming sub-sections.

### 3.1. Visually Improved Radar Composites

Two examples of radar composites with and without the quality information are presented in Figure 4. The panel shows the composite of seven (nine) radars in the left (right) column (each corresponding to a different weather event), first without the quality information (top), then with the QI algorithm described within this paper (center), and finally, a composite field of QI (bottom) is presented. The difference between the first two pairs of radar composites is clear. The disturbing, non-meteorological echoes, mostly caused due to local wi-fi routers with their frequencies interfering with the radar signals, were removed. Consequently, the radar composite with the quality information is evidently cleaner and offers a more reliable overview of the current meteorological situation.

In the examples presented in Figure 4, the overall QI was constructed on the basis of all specific QIs but the one based on the beam height above the terrain since its usage contradicts with the concept of the radar product CAPPI 2 km.

The visual improvement of the radar composites is particularly appreciated in aviation. Pilots, dispatchers, and air traffic specialists put value on being able to make decisions on the basis of 'clean', QI-based radar maps and not being forced to mentally filter out the unwanted information from the old radar composites with no quality information included. Thus, cleaner radar images enhance aviation safety; they allow for the more convenient and precise planning of flight routes as well as the necessary maneuvering of air traffic in real time operations in situations of the quick development of dangerous weather phenomena.

A QI-based radar composite that is updated in real time is an excellent tool as a separate layer on the air traffic management radar display at an ATC (air traffic control) workplace. A radar display that has the ability to measure distances by mouse and integrated radar data as a background represent invaluable support compared to separate radar displays at an ATC workplace, for instance to meet the regulations that an aircraft should fly at least 10 km from a cumulonimbus cloud, etc. In addition to this, clean radar composites do not purely offer visual advantages: a higher quality radar image has the potential to produce fewer false alarms by automatic warning systems.

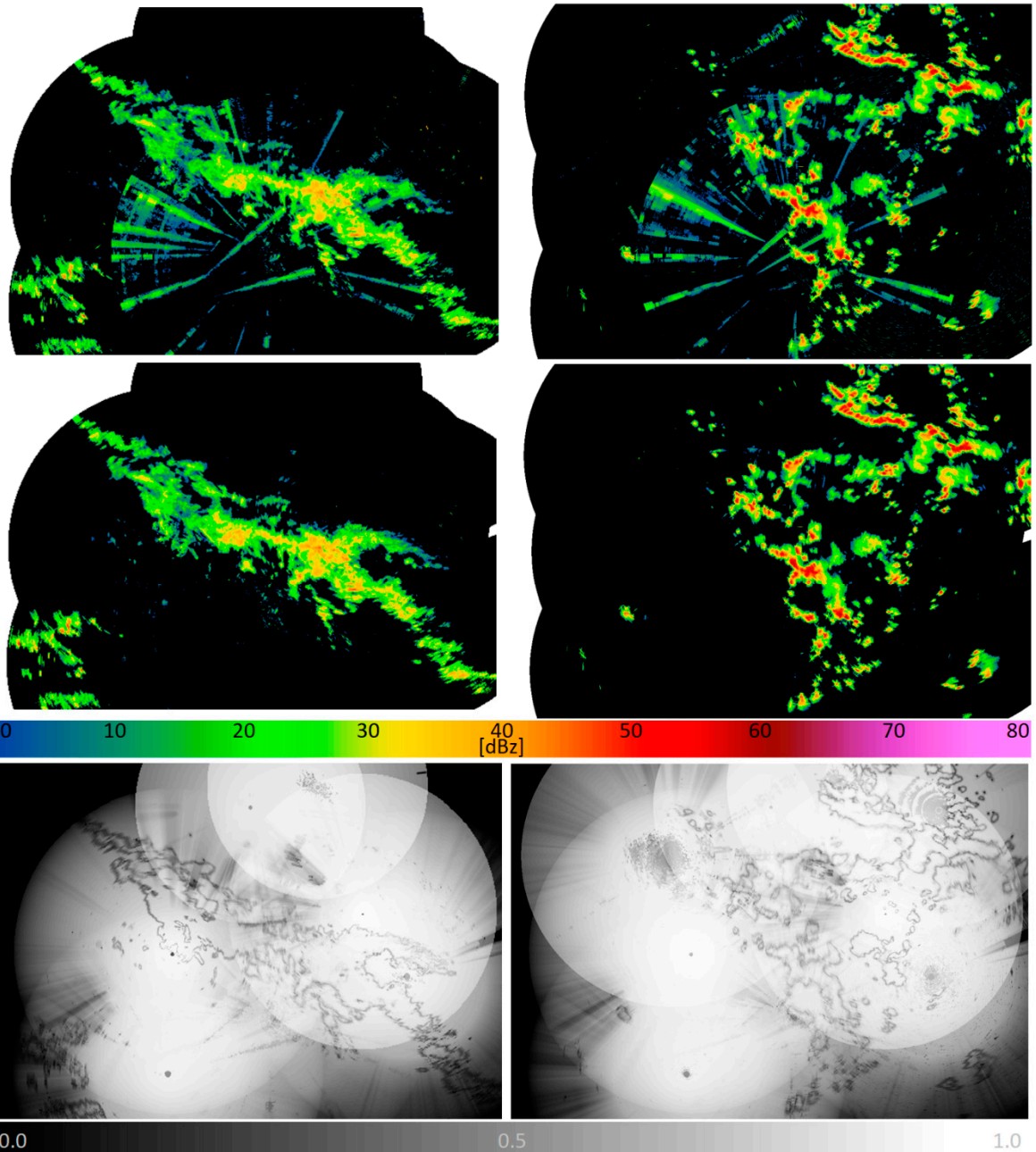

**Figure 4.** Examples of visual improvements in a composite of several radars from the Central Europe (Slovakia, Hungary, Poland and Czech Republic). The column on the right (left) demonstrates a weather event from 5 August 2015, 13:10 UTC (19 August 2015, 12:30 UTC), by means of a composite of seven (nine) radars. The composites at the top are without quality information, whereas in the center, the quality information is incorporated in the process of compositing. The bottom line presents the corresponding fields of the final QIs. The CAPPI 2 km radar product was used: constant altitude PPI, i.e., a cross-section across a number of different PPI scans at the pre-defined altitude of 2 km above the ground.

### 3.2. QPE Improvement

3.2.1. Rainfall Mapping

The *qPrec* software for quantitative precipitation estimation was developed in an iterative approach. Each upgrade or change of the software (e.g., incorporation of an additional QI, new input field, and modified algorithm) was validated against the 24-h precipitation amount from the network of about 600 climatological and pluviometric stations of the SHMU. A gradual expansion of the set of QIs in the *qRad* software is mani-

fested in the enhanced precision of the 24-h precipitation estimation by the *qPrec* software. This fact is illustrated in Figure 5.

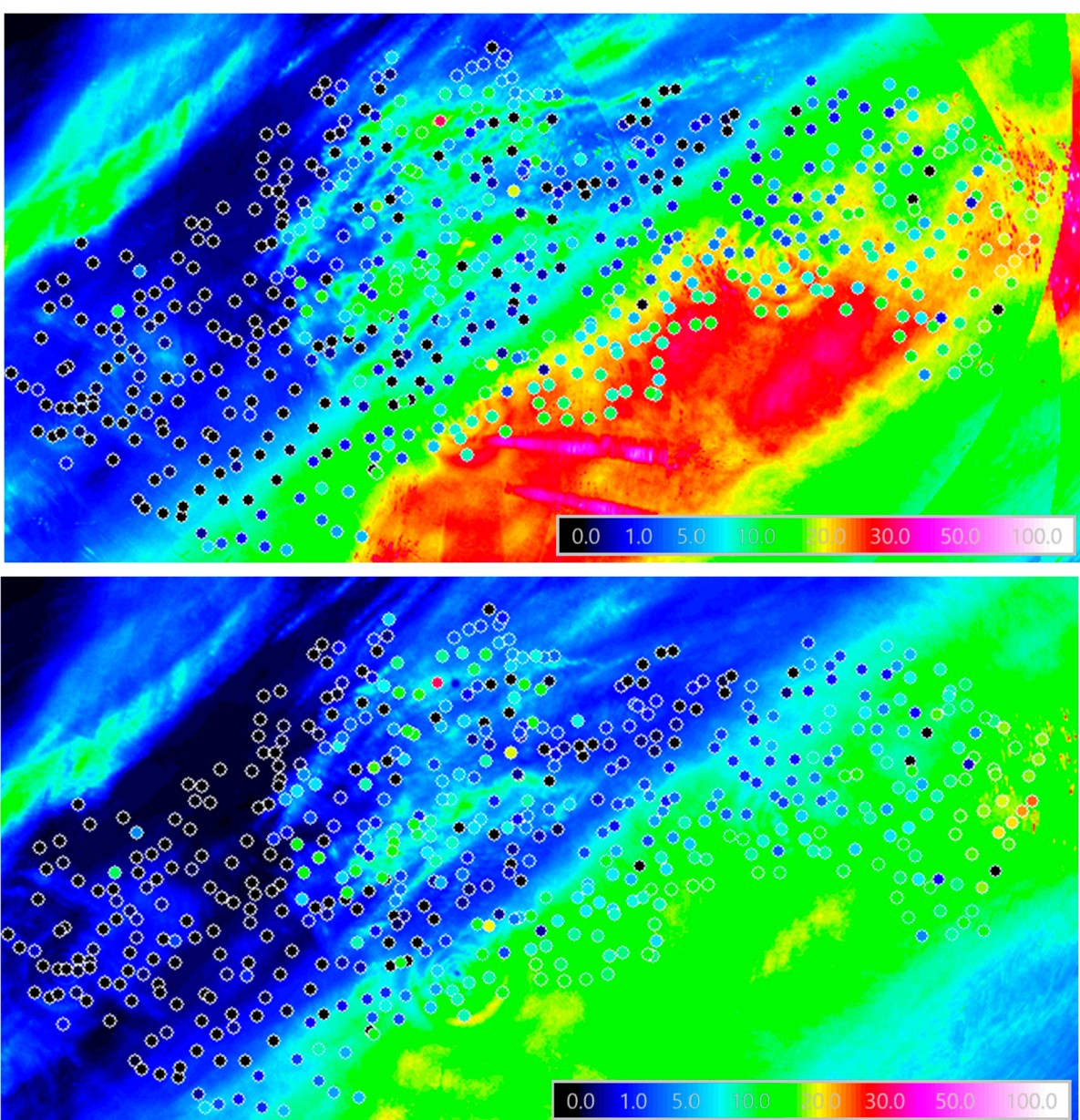

**Figure 5.** A comparison of the 24-h rainfall measured with the network of ~600 climatological and pluviometric stations in Slovakia (octagons) on 30 June 2017, with the estimation of those based on the radar data by means of a simple Marshall–Palmer relationship without quality information (**top**) and to the estimation of those by the *qPrec* software with the calibrated reflectivity-rainfall-rate relation and quality information (**bottom**).

Here, a traditional method of precipitation estimation based on the simple Marshall–Palmer relationship with no quality information (Figure 5 top) is compared with that derived by the *qPrec* software, making use of the QIs (Figure 5 bottom), whereas the 'true' information on the precipitation amounts from the pluviometric stations are shown in both figures. The Marshall–Palmer conversion relationship between the reflectivity factor $Z$ [mm$^6$/m$^3$] and the rainfall rate $R$ [mm/h] was used in its simplest form:

$$Z = 200R^{1.6} \tag{8}$$

There were significant differences in the radar-based estimation of the precipitation field (based on a composite of four radars located in Slovakia), mostly at the southern borders of Central and Eastern Slovakia, and in the Eastern parts of the country where the traditional Marshall–Palmer-based approach clearly overestimated the observed precipitation amounts. The QI-based estimation of the precipitation field, on the other hand, was much smoother, and successfully removed the majority of the non-meteorological signals as well as non-realistic estimates due to the bright band.

A similar analysis is presented in Figure 6; however, instead of a 24-h accumulation, the focus is set on a long-term period, showing the comparison of the observed vs. estimated rainfall amounts for a 5-month period (1 May 2017–30 September 2017). The colored dots in the maps indicate the ratio $r_{RG}$ of the radar-based estimation of rainfall amounts and the observed ones for three different approaches. A summary information on the frequencies of occurrence of the given values of $r_{RG}$ is presented in Table 1.

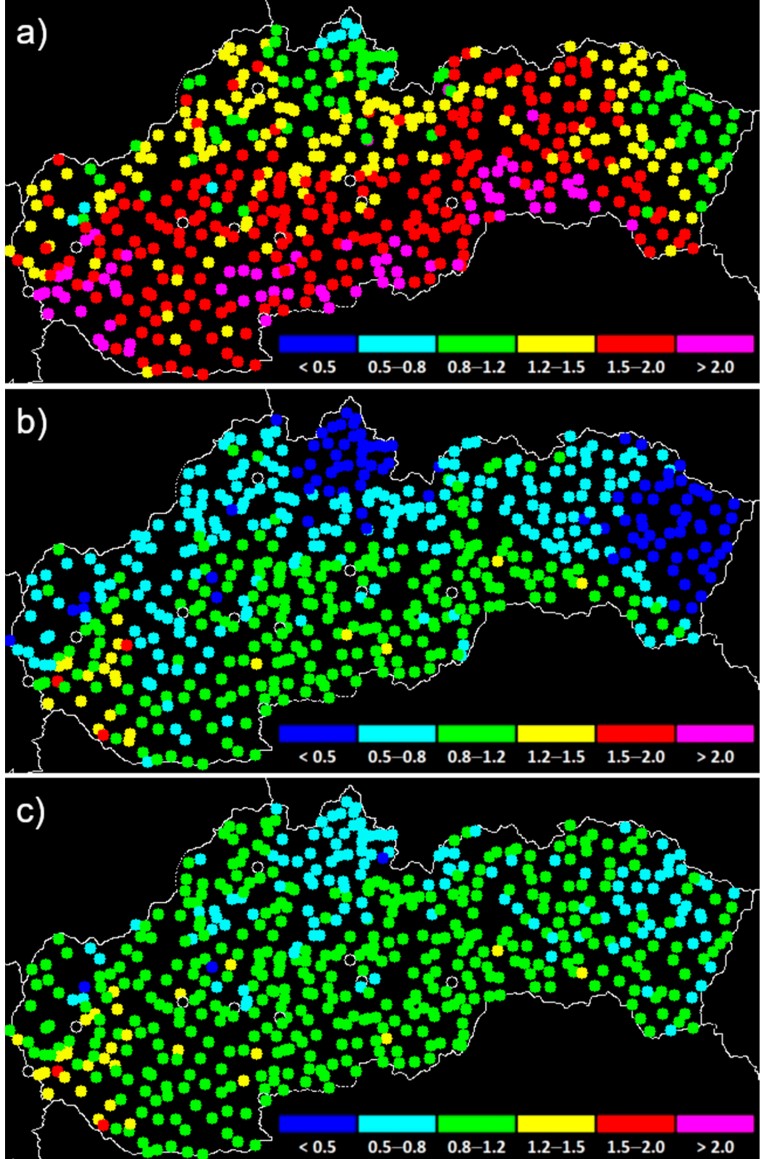

**Figure 6.** Ratios of the radar-based precipitation and the observed precipitation amounts for a long-term period (5 months, May to September, 2017) in Slovakia. The radar-based precipitation was estimated (**a**) using the Marshall–Palmer relationship, (**b**) by means of a calibration, but with no quality information, and (**c**) by means of a calibration, and with the quality information included. The analysis was based on 570 stations.

**Table 1.** Summary of the frequency of the occurrence of sites (in %) with a given radar/gauge ratio $r_{RG}$ in three maps of Figure 6. Based on the network of 570 pluviometric stations.

| Radar/Gauge Ratio $r_{RG}$ [−] | Marshall–Palmer [%] | Calibration, no QI [%] | Calibration, with QI [%] |
|:---:|:---:|:---:|:---:|
| $r_{RG} < 0.5$ | 0.0 | 17.1 | 0.5 |
| $0.5 < r_{RG} \leq 0.8$ | 1.9 | 38.5 | 22.4 |
| $0.8 < r_{RG} \leq 1.2$ | 14.9 | 40.2 | 71.6 |
| $1.2 < r_{RG} \leq 1.5$ | 27.9 | 3.7 | 5.1 |
| $1.5 < r_{RG} \leq 2.0$ | 42.5 | 0.5 | 0.4 |
| $r_{RG} > 2.0$ | 12.8 | 0.0 | 0.0 |

In the first map (Figure 6a), the rainfall amounts were estimated on the basis of the Marshall–Palmer relationship (Equation (8)) The map indicates that the simple Marshall–Palmer-based approach resulted in overestimation of the true precipitation amounts, at approximately 85% of the stations. The highest rates of the overestimation (with the radar/gauge ratio $r_{RG}$ exceeding 2.0) can be found in the southern parts of the country where lowland character dominates. Ratios with the most acceptable values ($0.8 < r_{RG} \leq 1.2$) as well as underestimation can be found at a considerably smaller number of sites (Table 1).

A completely different picture of the radar/gauge ratios can be found in the second map (Figure 6b), where the radar-based rainfall was estimated by means of a calibration using the true rainfall measurements from the rain gauge network, but with no quality information included. For the calibration, a completely different data set was used. Here, 40% of the sites are associated with an acceptable radar/gauge ratio. The most remarkable drawback of this figure is that, at more than half of the sites (predominantly in the NW, N, and NE, i.e., generally in the hilly and mountainous parts of the country), the true rainfall is underestimated. The number of cases with overestimation is practically negligible.

Finally, the third map (Figure 6c) shows the results of a similar analysis where both the calibration and the quality information were used in the procedure of the radar-based rainfall estimation. The dominance of the green color indicating the most acceptable radar/gauge ratios (~72% of all cases) justifies the added value of the QIs in the QPE (Table 1). The percentage of sites with over- and under-estimated rainfall is the lowest among the three different approaches.

### 3.2.2. Hydrological Modelling

As mentioned before, one of the most effective ways to evaluate the performance of an improved QI-based QPE is to verify it through different weather forecasting, nowcasting, or hydrological models. We also carried out such a test, by feeding the outputs of the *qPrec* software as inputs in the hydrological model HBV (Hydrologiska Byråns Vattenbalans, Hydrological Bureau's Water Balance [30]), which is, among others, operationally used by the hydrological forecasters of the SHMU and hydrological experts of MicroStep-MIS. Figure 7 presents a comparison of three series of daily discharges for a selected 12-month period (1 November 2016 to 1 November 2017) for the Hron River at the gauging station Polomka (630 m a.s.l., Central Slovakia; the area of the upstream basin 329.54 km$^2$) as the observed one, and two further series as a result of hydrological modelling on the precipitation estimation by the *qPrec* software and the INCA model (Integrated Nowcasting through Comprehensive Analysis [31]), respectively.

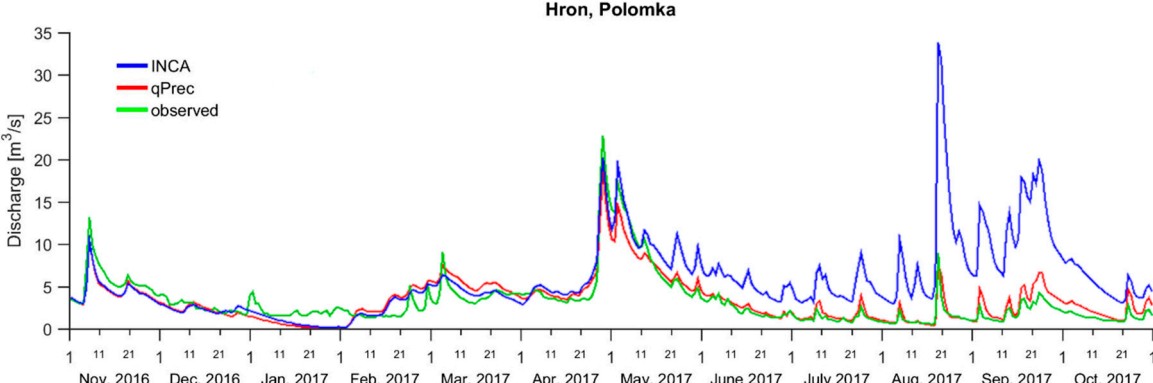

**Figure 7.** Course of the discharge values with an hourly time step for the Hron River at Polomka (Central Slovakia) from 1 November 2016 to 1 November 2017. The observed values are displayed in green, whereas the red (blue) color indicates the estimated discharge from the *qPrec* software (model INCA—Integrated Nowcasting through Comprehensive Analysis [31]).

In the hilly and mountainous regions of Slovakia (such as the Hron River at Polomka), the largest discharges are observed either due to snowmelt during the late spring months (April–May) or due to convective rainfall or stratiform precipitation during the summer or autumn months. Both effects are clearly discernible in Figure 7: (i) the increased discharge values due to the heavy snowmelt around May 1 where the corresponding flood wave attenuated slowly and lasted for a couple of weeks, and (ii) the shorter and quicker flood events in the warm half year. Figure 7 further shows that the outputs of the hydrological model reflect these phenomena in a different way. Both the *qPrec*- and the INCA-based results were able to follow the course of the observed discharge (also with the influence of partial snowmelt) during the cold half year acceptably well. Significant differences in the model behavior appeared in the warm half year where the connection between the precipitation and the discharge was more straightforward. The *qPrec*-based hydrograph copied the behavior of the observed discharges both in terms of timing and magnitude (with minor underestimation), whereas the INCA-based outputs constantly and significantly overestimated the real-world values.

The bar plot in Figure 8 explains the findings derived from the hydrographs. The differences in the outputs of hydrological modelling stem in discernible differences in the time series of 1-h area-averaged rainfall in the upstream basin: the INCA-based rainfall estimates generally exceed the *qPrec* values, particularly in the period starting from May. Due to the fact that there is only a single meteorological observatory in the given basin, a comparison of the INCA and *qPrec* estimates with the gauge data is omitted.

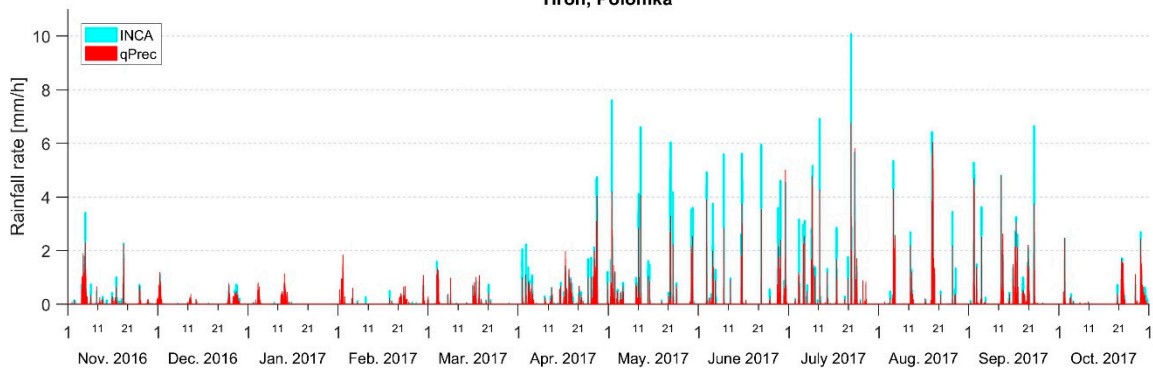

**Figure 8.** Time series of 1-h area-averaged rainfall rates (INCA vs. *qPrec* estimates) in the Polomka basin at the Hron River, from 1 November 2016 to 1 November 2017.

It is beyond the scope of this paper, to find the reason for this behavior of the INCA model. However, that the hydrological model based on the *qPrec* product results in a more precise estimation of the observed discharge than that based on the INCA precipitation analyses is important.

### 3.2.3. Runway Condition

Section 3.1 demonstrated the practical applicability of the *qRad*-based radar composites for pilots and air traffic managers. In addition to the *qRad* software, the *qPrec* package also offers clear benefits for aviation, namely, through an estimation of the precipitation amounts (QPE) directly on the runways.

Assessment of the runway condition and optimization of the airport operation management are key issues in the research of air traffic management, particularly in the SESAR (Single Europe Sky Air Traffic Management Research) program [32]. The amount of water, regardless of its phase, is a crucial factor for the safety of aircraft landing. On a wet or contaminated runway, the risk of adverse effects on the plane's braking performance increases. During the cold season, snow cover and/or ice may significantly decrease the plane's braking performance, whereas, during the warm season, the thickness of the water film on the runway affects the risk of aquaplaning.

In our experiment, we focused on the observation of the height of the water film on the runway surface and attempted to estimate this on the basis of radar measurements. The mini-experiment was only performed for the warm season (more precisely, the summer period of June/July/August), mostly for the following reasons. First, to perform a distant estimation of the amount of precipitation in a solid phase is a complicated task mostly due to the effect of a number of further physical and thermodynamic processes (e.g., melting, freezing, snow compacting, and wind effects). Second, climate change scenarios for the Central Europe generally foresee changes in the intra-annual distribution of the precipitation: one can expect more frequent and more intensive precipitation events in all seasons, whereas the mean precipitation is supposed to increase (decrease) in winter (in summer) [33–35].

The experiment was performed at the International Airport Poprad-Tatry (Slovakia; ICAO: LZTT; IATA: TAT). The location of the airport, with respect to the network of four meteorological radars of SHMU is depicted in Figure 9. This reveals that the coverage of the target area by meteorological radars is sufficient: the height of the layer that remains invisible for radars is approximately 300 m, which is acceptable from the point of view of detection of storms.

The height of the water film on the runway surface is measured at three locations of the runway (with a length of 2600 m and oriented in the 09/27 direction): near its both ends, and approximately in the middle of the runway. The intelligent road sensors Lufft IRS31Pro-UMB provide measurement of the water film height with a temporal resolution of 1 min.

On the other hand, the amount of precipitation on the runway was also estimated by means of two methods of QPE on the basis of the CMAX radar product: in a traditional way, using a simple Marshall–Palmer relationship (Equation (8)) with no QI included (termed simply as CMAX), and using the *qPrec* package with the QI included (termed as *qPrec*). Corresponding to the location of each sensor, the maximum precipitation amount was estimated approximately within a 1-km radius and within an X-minute time window (where $X \geq 4$). Finally, for each datum, the maximum of the three maximum values was selected for the analysis. We propose that the selected options (the spatial and temporal resolution and the method of maxima) ensures that the QPE can capture the rainfall even in the case when the core of the rainstorm does not hit exactly the runway.

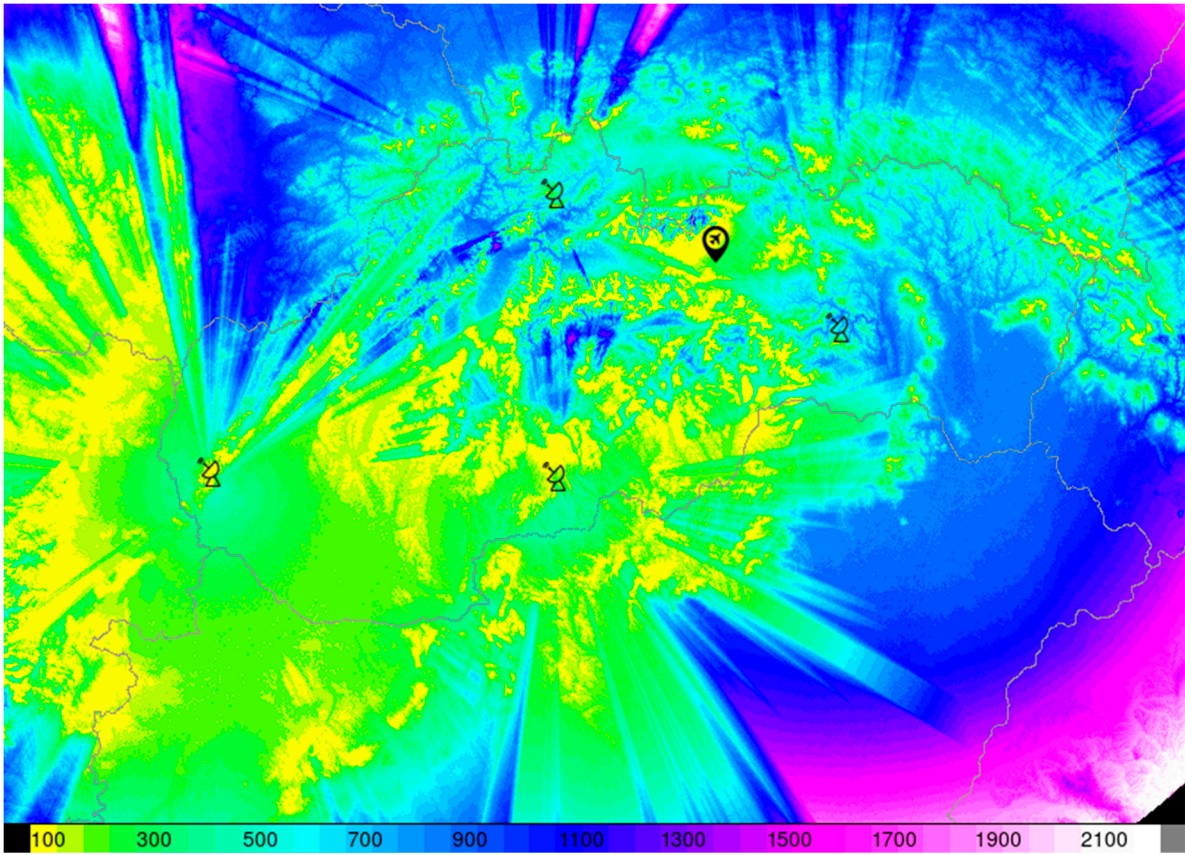

**Figure 9.** Coverage of the territory of Slovakia by the network of four meteorological radars of SHMU, expressed in terms of the height of the lowest visible radar beam at the given point (in m above the ground). The location of the Poprad Airport is denoted by the icon of a plane. The radar icons denote the locations of Malý Javorník (West), Kubínska hoľa (North), Kojšovská hoľa (East), and Španí laz (South).

The results of the analysis are demonstrated in Figure 10. This presents the scatter plots of the measured water film heights vs. the estimated rainfall amounts at the runway for the summer months of the years 2019 and 2020 for both of the QPE methods considered and for the rainfall accumulation time of 4 min.

The linear relationships between the water film heights and the rainfall amounts corresponding to the particular QPE method were weak in both cases. The traditional approach with no QI was characterized by poorly estimated rainfall amounts that often appeared as evident outliers—in some cases exceeding values of 10 mm. These are striking mostly in the right bottom regions of the plot, where they are related to very low values of the measured water film heights. The large spread of the QPE estimates was, therefore, reflected in a very low coefficient of determination ($R^2 = 0.18$). On the other hand, the cloud of points for the *qPrec* estimates was more compact. This was underpinned by the fact that the estimated 4-min rainfall amounts did not exceed 3 mm. While the coefficient of determination ($R^2 = 0.36$) of the *qPrec* method was still relatively low, it was sufficient to derive the overall conclusion of the experiment, i.e., the added value of the novel, QI-based approach to QPE.

The results clearly depended on the settings of the experiment, particularly on the time window of the accumulation of the *qPrec* amounts. The *qPrec*-based scatter plot indicated that the estimates of the 4-min amounts underestimated the observed heights of the water film. The 5- and 6-min time windows yielded similar scatter plots (with the $R^2$ practically with the same magnitude); however, the corresponding linear regressions generally indicated overestimation of the observed values (not shown).

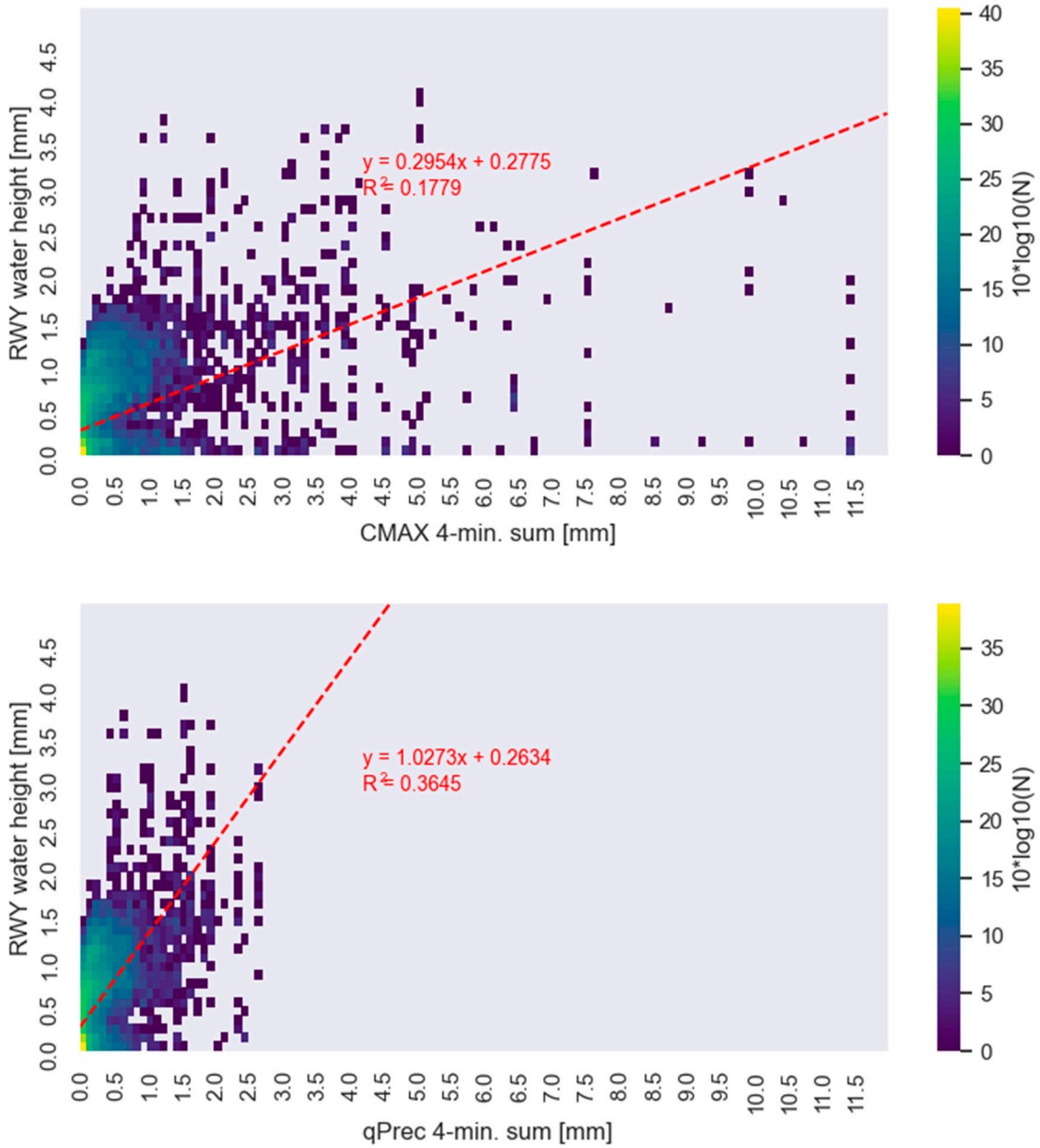

**Figure 10.** Scatter plots of the measured water film height on the runway (RWY) and the 4-min QPE for two summer periods (June/July/August) of the years 2019 and 2020 at the Poprad Airport, Slovakia. Top: traditional QPE, based on a simple Marshall–Palmer relationship, with no quality information included (CMAX). Bottom: novel QPE, based on the *qPrec* package. The logarithmic color scale indicates the frequency of occurrence ($10 \times \log_{10}(N)$) of the pairs of the values in the bins with a width of 0.1.

## 4. Discussion

Radar composites of high quality are not only needed in the international exchange of radar data but also locally in geographically complex terrain. In such cases, a locally installed smaller meteorological radar (such as an X-band radar, [36–38]) could supplement the information on the lower layers of the atmosphere that remain invisible for other radars. Moreover, with their higher temporal and spatial resolution, these can contribute to a more detailed characterization of precipitation systems, particularly in the case of detection and nowcasting of severe weather phenomena [38]. One of the most important drawbacks of the X-band radars is their limited maximum range (80–100 km); however, this disadvantage is balanced by the network of C-band radars with a much wider spatial

coverage. Therefore, developing correct methodologies for creating composites based on a hybrid network of X-band and C-band radars is of a great importance [39,40].

The quality-based approach to the correction of the radar data offers a high degree of flexibility for the researchers. The selection of individual QIs, their definition (e.g., whether one uses linear or exponential relationships, as well as the selection thresholds and constants), the method of scaling, and the technique of combination are rather subjective. There is not a unique, objective, and standardized way of making use the QIs themselves. Their benefits depends on the user's application and on the particular method of their implementation (weighting, thresholding, using the maximum of the QIs, etc.).

The effect of various QI algorithms are usually evaluated through different use-cases. For example, when the radar data are used for QPE, the benefits of the QI algorithm should be evaluated as the improvement of the QPE performance. Similarly, when the radar data are used in numerical weather prediction models, the QI algorithm should be evaluated by its influence on the model outputs. Nowcasting based on cleaner radar images should be more reliable, as the non-meteorological echoes—which can cause unrealistic motion vector patterns and subsequently wrong extrapolation of the rainfall field—are suppressed or filtered out.

As soon as the first versions of the *qRad* software became available for internal use, we experienced positive feedback from the forecasters and internal users of SHMU. They emphasized the quality of our visualization where the majority of non-meteorological and false echoes were removed or at least minimized by using the quality information. Therefore, when presenting interesting meteorological situations to the official channels of SHMU, they clearly preferred the *qRad* outputs over the products from the official software of the radar manufacturer.

The software *qPrec* offers a wide range of benefits from the point of view of climatology. The ground-based network of pluviographs can never be sufficiently dense to catch the high temporal and spatial variability of the local intensive thunderstorms. *qPrec* is able to detect these phenomena and allows for a more precise reconstruction of the meteorological conditions of their occurrence, evolution and decay. This is particularly important in the case of the evaluation of insurance issues and other expert opinions.

Hydrological validation is also a part of the iterative development process of the *qPrec* software. The help of hydrologists of SHMU is kindly appreciated in this way. In the operational forecasting of river discharges, the catchment precipitation was originally estimated on the basis of the outputs from the INCA model. This routine has been changed during the past couple of years as the catchment precipitation estimated by means of the *qPrec* software was found to yield more realistic results than when based on the original concept of INCA.

The results of the presented experiment with the water amount on the runway cannot be generalized at this stage. To do so, a more elaborate study is necessary, based on longer periods of observation and on a more complex (possibly non-linear) model with more predictors and with assumptions on the runoff of the water from the runway, the accumulation of the water on the runway from previous rainfalls, its evaporation, etc. The experiment indicated the potential for utilization of the QI-based QPE in aviation meteorology. Water levels on the runway can be practically estimated by the QI-based method at a multitude of airports within a given region, and consequently the pilot may have the chance of choosing to land at the airport with the least risky runway conditions. An even more elaborated model would, in principle, be able to cope with nowcasting of the runway conditions.

Both software packages are under development, with a many potentially useful ideas for their further improvement in the near future. In *qRad*, a fine tuning of the parametrization of the individual QIs is an ongoing process. Clearly, there is a possibility of extending the range of the involved QIs—currently, QIs based on bright-band detection and the radial components of wind data are in the testing phase. A further option to improve the *qRad* software is in taking advantage of specific characteristics of the dual

polarization radars [24]. We also plan to test the possibility of the incorporation of X-band radars into the composites of C-band radars, which is a challenging task.

We further intend to improve the *qPrec* software, particularly regarding its statistical background. First, the probability of precipitation should be calculated on the basis of the Bayesian approach. Secondly, the current version of the *qPrec* package is based on the hypothesis of normally distributed errors. This simplification should be revised in the future, and a more sophisticated model of error distribution should be implemented. More generally, the performance of the *qPrec* software will be tested using a wider spectrum of precipitation-related characteristics. Beyond the CMAX product that has formed the basis of the calibration of the precipitation field so far, further radar products (e.g., CAPPI) and/or precipitation-related variables (satellite data, lightning characteristics, etc.) will be involved and tested.

Both software packages are primarily intended for internal use at the SHMU. The *qRad* software is also being used by the Hungarian Meteorological Service in the framework of an H-SAF product validation. Beyond this, no commercial distribution of the products is expected.

## 5. Conclusions

In this paper, an attempt was made to present the philosophy and specific features of two software packages developed at the Radar Department of the Slovak Hydrometeorological Institute. Both the *qRad* and *qPrec* packages are based on the concept of quality information (QI) of the radar measurements. Different QIs express different factors (environmental, meteorological, technological, etc.) that influence the radar data and their quality. The overall quality of the individual radars can then be considered in the construction of radar composites, and consequently a wide spectrum of unwanted features can be suppressed or completely eliminated in the new, clean radar composites. The *qPrec* package also takes advantage of the radar quality information in the quantitative estimation of precipitation amounts and intensities.

We demonstrated the value of the software packages via different use-cases. These involved clean radar composites that may directly contribute to increased flight safety, high quality estimation and mapping of long-term precipitation, and improved performance of hydrological modelling as a result of the QI-based estimation of catchment precipitation.

Improved radar composites also offer indirect benefits in accordance with sustainability and environmental protection. These are, for instance, fuel savings in the case of effective planning or modification of flight routes based on the clean (QI-based) radar composites. A further, sustainability-compatible benefit of the QI-based QPE is the estimation of the risk of aquaplaning on runways. Having information on the runway surface conditions, pilots may avoid runway excursions and, thus, reduce the risk of ecological or other forms of disasters.

The results of this study present a positive answer to the research question, which is reflected in the created software tools *qRad* and *qPrec*, verified on various praxeological examples from various fields of hydrology and meteorology. The presented solutions provide innovative contributions for academic discussion and practice, particularly the strengthening of situational awareness to increase flight safety and to support sustainable and environmentally friendly aviation activities.

**Author Contributions:** Conceptualization, L.M. and M.J.; methodology, L.M. and M.J.; software, L.M.; validation, L.M.; formal analysis, M.G. (Martin Gažák) and M.G. (Martin Gera); investigation, L.G.; resources, M.J.; data curation, J.B.; writing—original draft preparation, L.G. and J.B.; writing— review and editing, L.G. and M.K.; visualization, L.M.; supervision, M.K. and M.G. (Martin Gažák); project administration, M.G. (Martin Gažák) and M.G. (Martin Gera); funding acquisition, M.J. and J.B. All authors have read and agreed to the published version of the manuscript.

**Funding:** This research was funded by the project "System of More Accurate Prediction of Convective Precipitation over the Regional Territorial Unit" (No. VI20192022134, Ministry of Interior of the Czech Republic), and partially funded by SESAR JU, Grant Agreement No. 874472, under EU's Horizon

2020 Research and Innovation Program. This publication is also partially a result of the project implementation "Scientific Support of Climate Change Adaptation in Agriculture and Mitigation of Soil Degradation" (ITMS2014+ 313011W580) supported by the Integrated Infrastructure Operational Programme funded by the ERDF.

**Institutional Review Board Statement:** Not applicable.

**Informed Consent Statement:** Not applicable.

**Acknowledgments:** The contribution of hydrologists of the Slovak Hydrometeorological Institute is acknowledged.

**Conflicts of Interest:** The authors declare no conflict of interest.

## Appendix A. Specific Measures of the Radar Data Quality

### Appendix A.1. Distance from the Radar

Each radar bin was evaluated according to its distance from the radar. The radar beam broadens and its volume grows as it propagates in the atmosphere. The resulting measurement, then, represents an average of a larger volume. In this case, a linear relationship is used:

$$
qi_{1,i} = \begin{cases} 1.0; & r < r_{min} \\ 1.0 - \frac{r - r_{min}}{r_{max} - r_{min}}; & r_{min} \leq r \leq r_{max} \\ 0.0; & r > r_{max} \end{cases} \tag{A1}
$$

where $r$ is the distance from the radar and $r_{min}$ and $r_{max}$ are parameters of the computation (with default values of 0.0 and the maximum range of the radar, respectively). The index $i$ indicates that the quality index is estimated for the $i$-th available radar (the same holds true for the upcoming equations).

This quality index (QI) belongs to the most elementary ones in assessing the radar signal quality: it appeared in the same [19–21] or in a similar form with most studies (such as using the exponential relationship [16]). There may be sophisticated methods in defining the minimum or the maximum ranges of the radar [19–21]. Beyond this, Fornasiero et al. [16] incorporated a correction factor that depended on the distance of the target bin from the nearest radar.

### Appendix A.2. Beam Height above the Terrain

Some quality factors only influence certain types of radar products. The beam height above the terrain should be useful especially for rainfall estimating products—the estimated rainfall is expected to be more precise when the target bin is near the surface of the terrain. The related QI is computed as follows:

$$
qi_{2,i} = \begin{cases} 0; & h < 0 \\ 1 - \frac{h}{h_{max}}; & 0 \leq h \leq h_{max} \\ 0; & h > h_{max} \end{cases} \tag{A2}
$$

where $h$ is the bin height above the terrain and $h_{max}$ is the user defined maximum height above the terrain. Note that the case $h < 0$ occurs when the radar beam hits the ground, i.e., the projected radar beam is under the surface.

The quality index $qi_{2,i}$ also belongs to the fundamental ones, implemented for instance by Tabary et al. [18]. Szturc et al. [20,21] adopted this concept in a slightly different way: they preferred working with the height of the lowest radar scan, which, for a given radar represents a constant value, depending on the actual altitudes of the surrounding terrain.

### Appendix A.3. Beam Blockage by the Terrain

The percentage of the transmitted energy blocked by the terrain was computed according to the energy distribution along across the radar beam. The 10-m resolution ASTER

GDEM (Advanced Spaceborne Thermal Emission and Reflection Radiometer Global Digital Elevation Model) digital terrain model was used in the computation. The resulting QI is then:

$$qi_{3,i} = \begin{cases} 1 - B; & B \leq B_{max} \\ 0; & B > B_{max} \end{cases} \tag{A3}$$

where $B$ is the actual blockage and $B_{max}$ is a user-defined threshold of blocking, both from the interval 0 to 1. The software offers the possibility to correct the blockage by dividing the measured energy by the blockage.

The issue of blocking the radar beam is rather complex, as there are a number of objects (beyond clouds and hydrometeors, there is the relief, birds, insects, etc.) that interact with the radar signal in different ways. Therefore, different authors presented different approaches to the mathematical quantification of these effects. For instance, Friedrich et al. [19] defined a single empirical form for the effect of beam shielding that incorporated the ground clutter. On the other hand, both Tabary et al. [18] and Fornasiero et al. [16] treated the beam blocking and the ground clutter as two independent effects, defining the latter one in a binary way: 0 if ground clutter was present, and 1 otherwise. The weakening character of the radar signal along the radar beam was further described, for instance, by the attenuation by the hydrometeors [19], the occultation rate [18], and so-called path integrated attenuation [17].

*Appendix A.4. Similarity to the Surrounding Bins*

In the *qRad* software, two bins are considered similar when their values differ by less than a given threshold. The similarity QI is, then, the percentage of the similar bins within a defined window around the given bin. The default setting is a $3 \times 3$ window centered on the target radar bin; thus, its value is compared with those from the other eight surrounding bins (this setting is customizable by the user). This QI is aimed at finding spikes or smaller holes and to evaluate the bins on the borders of the detected clouds. The software enables replacement of the value of the bin with a QI lower than a defined threshold with a QI-weighted average of the surrounding values.

A similar philosophy was applied by Szturc et al. [20,21] who assessed the spatial variability of the rain field by estimating the variance of the rainfall rate within a window of $3 \times 3$ or $5 \times 5$ centered on the target radar bin.

*Appendix A.5. Time Quality Index*

The time of the measurement is evaluated in this QI. More precisely, this QI evaluates the temporal degradation of the radar signal within a single antenna rotation. This is computed as a linear relationship of the difference $\Delta T$ between the time of the measurement and the given product validity time:

$$qi_{5,i} = \begin{cases} 1 - \frac{\Delta T}{\Delta T_{max}}; & \Delta T \leq \Delta T_{max} \\ 0; & \Delta T > \Delta T_{max} \end{cases} \tag{A4}$$

where $\Delta T_{max}$ is a user-defined maximum allowed time difference.

This QI aims at quantifying the time interval that is needed to carry out a full PPI scan (plan position indicator: a complete $360°$ scan where the elevation angle of the radar beam is set to a constant value). The larger the $\Delta T$ (i.e., the slower the radar revolves), the more significant the observed change in the rainfall field may be at the start of the rotation and at the same position of the radar after a full $360°$ circle, respectively.

The temporal degradation of the radar signal is also considered in [17]. Szturc et al. [20,21] aimed to catch the temporal dimension of the rainfall field in a different way. They expressed the temporal variability of the rainfall by applying an *n*-hour moving window method both backward (i.e., on the past estimates of the rainfall field) and forward (i.e., on its forecasts).

### Appendix A.6. Cloud Type Quality Index

The Cloud-Type product of the NWC SAF (Nowcasting Satellite Application Facilities [25]) is used in this QI. Non-undetected bins are evaluated according to the cloud type in their position. The QI is associated according to Table A1.

**Table A1.** Quality index (QI) according to cloud type.

| Cloud Type | QI |
| --- | --- |
| Cloud-free (land, sea, snow, ... ) | 0.01 |
| Fractional | 0.1 |
| High semi-transparent thin | 0.1 |
| Other clouds | 1.0 |

The *qRad* software enables replacement of the low QI bins with an undetected value according to the user settings.

The incorporation of the cloud-type QI was motivated by the goal to identify the cloud-related radar echoes, and on the basis of this information, to exclude or significantly reduce the non-meteorological clear sky echoes, particularly the R-LAN (radio local area network) or wi-fi interferences, ground clutter, and biological echoes. The QI uses the Cloud Type product from the NWC SAF software package computed from the Rapid Scan Service data every 5 min at the SHMU. A parallax-correction algorithm was applied to geographically match the satellite and radar products.

### Appendix A.7. Cloud Top Height Index

This index uses the Cloud Top Temperature and Height product of the NWC SAF [25] to assign a low QI (0.01) to the non-undetected bins found above the cloud tops. The software enables the replacement of suspicious bins with an undetected value.

Similarly to the Cloud type QI, this QI enables the detection, monitoring, and filtering of non-meteorological clear sky echoes, particularly in cases when a given area is covered by low clouds and the Cloud type QI is not able to distinguish the false echoes above the cloud top. The NWC SAF Cloud Top Height was computed from the same 5-min Rapid Scan Service data, and the parallax correction algorithm was also applied the same way as for the Cloud type QI.

### Appendix A.8. Average Quality Index

This QI is computed as an average of the user-selected QIs from a couple of last measurements. This is supposed to be useful to detect ground-clutter or other permanent error sources that may only be discernible by means of an analysis of a longer sequence of radar measurements. In our application, the combination of the cloud type QI and cloud top height QI was used to define the average QI.

### Appendix A.9. Constant Quality Index

The quality factor in this case is the radar's position itself, its hardware, and settings. Measurements of reliable and properly set radars are evaluated as good quality ($qi_{9,i} = 1.0$), while the data of less reliable or less precise radar sites are labelled with lower QI values. This quality index is also useful when other quality indices are not computed (due to missing data, etc.). In this case, the missing quality index (indices) can be replaced with a constant number.

The definition of constant QI was motivated by the experience where the radars of a given composite are of a different overall quality; for instance, some are more influenced by wi-fi signals than others, or radars from a given country are set to generally lower temporal resolution than those from the neighboring country. Constant QI is, therefore, aimed at expressing the degree of the relative quality of the individual radars contributing to a composite, and is often assessed on the basis of subjective experience.

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
