# Peer review of "Improved Radar Composites and Enhanced Value of Meteorological Radar Data Using Different Quality Indices"

_sustainability, doi:10.3390/su13095285_

Round 1

Reviewer 1 Report

The aim of this paper is to highlight the benefits of using quality controlled radar composites (with respect to raw data) to obtain more reliable, usable information on meteorological events. The manuscript specifically refers to aviation safety, but the topic is relevant for many other applications.

The authors illustrate the basic principles of two software packages they have developed (qRad and qPrec) and show the improvements that such procedures can achieve on the resulting radar composites and QPE.

In my opinion the topic is relevant and the overall quality of the paper is fair; however, there are some important remarks that I would like to point out.

First of all, although one of the main goals of the paper is the presentation of the qRad and and qPrec software, no information is given on the availability, expected distribution, licensing policies etc. of such software.

Secondly, apart from the visual improvements, the performance of the proposed procedures is evaluated only for cumulative precipitation values (24 hours or seasonal), whereas weather radar data is particularly advantageous for monitoring intense precipitation events, that are also the more relevant for aviation safety. Also the hydrological modelling example is performed at daily time scale, while the evaluation in case of one ore more extreme events would be more interesting.

In general the results are presented in a quite didactic, rough way; although this may be appropriate for an introductory paper as this one, I think that an effort should be made to include more quantitative information.

Other remarks:

The authors generally refer to ‘radar composites’ without specifying, for example, the radar type (e.g C-band, X-band, etc.). I think a brief discussion on this should be added.

Row 187, equation (2): what is the meaning of h<0 ?

Row 366: H-SAF refers generically to the satellite application facility, that produces several rain rate products as well as other variables (soil moisture, snow, …). The authors should be more specific about which H-SAF product(s) they refer to.

Row 387 (and later): Sigma symbol is to be used for standard deviation, whereas sigma-squared is for variance.

Row 391, figure 3: I understand the image is just an example, but details about which event/calibration data it refers to should be shown.

Row 410-414: Please clarify the choice of using the quality indices to weight the standard deviation.

Row 441, figure 4: A proper legend/colorbar should be added. Also time and location of the events should be indicated.

Rows 455-499: It is not clear to me if the qRad composites were actually available to the pilots of the three flights landing at Budapest airport or if the events described are just a reconstruction. In the latter case (retrospect reconstruction), the authors should describe briefly the expected advantages of having the qRad maps available, for example: could some decisions about flight route be taken earlier? Or in a different, more efficient, way?

Row 501, figure 5: A scalebar could be useful to have an idea of distances. Legend/colorbar should also be added.

Row 529, figure 6: If I understand correctly, zero rainfall and rainfall > 100 mm/h are both depicted in black. This may cause misunderstanding. Also, an overlay map of country borders could be useful.

Rows 580-581: please add information on the area of the upstream basin.

Rows 606, figure 8: I think a plot with comparison of time series of area-averaged rainfall in the upstream basin from gauges, INCA and qPrec (in addition to the comparison of derived discharge time series) should be shown.

Author Response

The response of our research team has been composed to both of the reviewers in the form of a single Word document.

Reviewer 2 Report

The article brings a lot in terms of the proposed solutions, but I believe that the research presented and the way of data presentation do not match the Sustainability journal. Rather, it should be submitted to Applay Sciences, etc.
The article is too long. Especially in the methodological section. The very technical description makes it hard to understand.
Abstract section: it should introduce the research topic in terms of environmental and public health goals, as it stands it is very technical. It does not encourage you to continue reading.
The introduction section: describes the technical part too much, little focus on the problem that the authors are solving in a non-technical aspect. Here should be a broad description of what the software will do for the environment and publch health. The authors wrote only one-two sentence about it.
It describes extensively what is in each section of the article, but there is no clearly defined purpose of the research.
Methodological section: too expanded, not very interesting, very technical.
Discussion section: should refer to the results, impact on the society, environmental and compare with the achievements so far in this field.
Conclusione section: too stretched, some of the content should be included in the Introduction and Disscusion sections, especially in terms of the potential environmental and public health impacts of the results reported.

Author Response

(The authors gave the same response as above.)

Round 2

Reviewer 2 Report

The authors responded to all comments from reviewers in the cover letter. However, in the text of the article, it is difficult to notice the changes made, because the authors did not mark them. Please, mark all added or removed elements with color.

Author Response

Please find attached the MS Word document with the Track Changes function, where all the modifications are highlighted. I hope it will help.
